# X-Linked Epilepsies: A Narrative Review

**DOI:** 10.3390/ijms25074110

**Published:** 2024-04-08

**Authors:** Pia Bernardo, Claudia Cuccurullo, Marica Rubino, Gabriella De Vita, Gaetano Terrone, Leonilda Bilo, Antonietta Coppola

**Affiliations:** 1Pediatric Psychiatry and Neurology Unit, Department of Neurosciences, Santobono-Pausilipon Children’s Hospital, 80129 Naples, Italy; 2Neurology and Stroke Unit, Ospedale del Mare Hospital, ASL Napoli 1 Centro, 80147 Naples, Italy; c.cuccurullo93@gmail.com; 3Department of Neurosciences, Reproductive Sciences and Odontostomatology, University Federico II of Naples, 80131 Naples, Italyleda.bilo@gmail.com (L.B.); 4Department of Molecular Medicine and Medical Biotechnology, University of Naples Federico II, 80131 Naples, Italy; gabriella.devita@unina.it; 5Child Neuropsychiatry Units, Department of Translational Medical Sciences, University Federico II of Naples, 80131 Naples, Italy; gaetano.terrone@unina.it

**Keywords:** X-linked, epilepsy, developmental and epileptic encephalopathies (DEEs), genetics

## Abstract

X-linked epilepsies are a heterogeneous group of epileptic conditions, which often overlap with X-linked intellectual disability. To date, various X-linked genes responsible for epilepsy syndromes and/or developmental and epileptic encephalopathies have been recognized. The electro-clinical phenotype is well described for some genes in which epilepsy represents the core symptom, while less phenotypic details have been reported for other recently identified genes. In this review, we comprehensively describe the main features of both X-linked epileptic syndromes thoroughly characterized to date (*PCDH19*-related DEE, *CDKL5*-related DEE, *MECP2*-related disorders), forms of epilepsy related to X-linked neuronal migration disorders (e.g., *ARX*, *DCX*, *FLNA*) and DEEs associated with recently recognized genes (e.g., *SLC9A6*, *SLC35A2*, *SYN1*, *ARHGEF9*, *ATP6AP2*, *IQSEC2*, *NEXMIF*, *PIGA*, *ALG13*, *FGF13*, *GRIA3*, *SMC1A*). It is often difficult to suspect an X-linked mode of transmission in an epilepsy syndrome. Indeed, different models of X-linked inheritance and modifying factors, including epigenetic regulation and X-chromosome inactivation in females, may further complicate genotype–phenotype correlations. The purpose of this work is to provide an extensive and updated narrative review of X-linked epilepsies. This review could support clinicians in the genetic diagnosis and treatment of patients with epilepsy featuring X-linked inheritance.

## 1. Introduction

X-linked epilepsies (XLE) can be considered as an expanding group of genetically heterogeneous and clinically variable conditions, with frequent coexistence of intellectual disability (ID) [1,2]. Particularly, epilepsy can occur as an isolated X-linked trait, as a component of recognizable X-linked syndromes or X-linked developmental and epileptic encephalopathies (DEEs). Moreover, the phenotype of X-linked epilepsy is complex, because of the variability of clinical manifestations that may be attributed to several factors, including the type of genetic mutation, epigenetic regulation, X chromosome random inactivation, and mosaic distribution.

The following is a comprehensive review of X-linked disorders and conditions associated with epilepsy, where epilepsy represents the “core” of the disease. Several important genetic pathways for both known disease genes and novel emerging X-linked genes are described and are discussed with relevance to clinical characteristics. This review might support clinicians in the genetic diagnosis of epileptic syndromes or DEEs when an X-linked mode of transmission is suspected.

## 2. Materials and Methods

A systematic search on MEDLINE PubMed was performed with the terms “X-linked”, “Epilepsy”, “Developmental and epileptic encephalopathy”, and “Seizures” (last search 31 December 2023). All manuscripts including cases with electro-clinical information were reviewed.

## 3. General Notes on X-Linked Disease Inheritance

The majority of disorders caused by mutations in genes on the X chromosome are inherited through the maternal germline and their phenotype depends on the sex of the affected individual [3]. The inheritance pattern of X-linked mutations can be dominant or recessive. The former is defined as X-linked dominant vertical transmission in which daughters of affected males are always affected, while the latter is defined as X-linked recessive vertical transmission in which carrier females pass the trait to affected sons (Figure 1) [3]. Females with X-linked dominant disorders typically have variable severity of their phenotype, a feature that is thought to be dependent on the X inactivation pattern [3]. X chromosome inactivation (XCI) is a process by which the dosage of proteins transcribed from genes on the X chromosome is equalized between males (XY) and females (XX) through the silencing of most genes on one of the two X chromosomes in females. Mammalian XCI results in the random silencing of one of the two X chromosomes in females early in development in order to ensure a balance in the dosage of X-linked gene expression between the sexes [4,5]. As the descendants of each cell keep the same pattern of inactivation, an individual heterozygote for an X-linked disease will be a mosaic, with two cell populations, one of which will express the normal and the other one the abnormal X chromosome. Consequently, some disorders demonstrate ‘mosaic’ or ‘patchy’ symptoms in heterozygous females [3].

## 4. Known X-Linked Developmental and Epileptic Encephalopathies

We reviewed the main electro-clinical features and the known genotype–phenotype correlations, including X-liked genes long known to cause DEEs, genes responsible for genetic and structural epilepsy from X-linked Neuronal Migration disorders and X-linked DEE genes that have emerged in the last few years. Our findings are summarized in Table 1 [6,7,8,9,10,11,12,13,14,15,16,17,18,19,20,21,22,23,24,25,26,27,28,29,30,31,32,33,34,35,36,37,38,39,40,41,42,43,44,45,46,47,48,49,50,51,52,53]. In Appendix A, we included XLE patients personally evaluated and treated (Appendix A).

### 4.1. PCDH19

Mutations in the *PCDH19* gene were originally identified by Dibbens et al. [54] in patients with epilepsy and ID limited to females (or Epilepsy, Female-restricted, with Mental Retardation, EFMR; OMIM# 300088).

#### 4.1.1. Genetic Features and Inheritance

Although the *PCDH19* gene is located on Xq22, this condition has an unusual X-linked mode of inheritance, sparing transmitting males, who are affected only when somatic mosaicism for the *PCDH19* mutation occurs. Males with cellular mosaics for the *PCDH19* gene with a similar clinical profile to that of affected females have been described, thus challenging the dogma that this is a disorder limited exclusively to females [55]. Mosaic male patients have the same phenotype as that presented by female patients, as reported in the literature, because they have the same hypothesized pathological mechanism of molecular interference [56].

#### 4.1.2. Epileptic Phenotypes

The hallmark feature of *PCDH19*-associated epilepsy is that seizures occur in clusters, lasting hours to days, often induced by fever, with onset around the first year of life [9]. A bi-phasic course with several brief clusters of febrile seizures followed by clusters of afebrile seizures has been described. Chemaly et al. proposed that fever sensitivity may be highest after age two years rather than at the onset of seizures [57]. The occurrence of a cluster of seizures is a typical feature, hence the name *PCDH19*-Girls Clustering Epilepsy (GCE).

Epilepsy is characterized by predominantly focal seizures, with affective symptoms (“fearful expression”) that emerged as a characteristic sign of seizures in this syndrome [6]. Hypo-motor seizures with staring and behavioral arrest are also frequent. Although focal onset is a constant feature, generalized seizures (such as tonic, myoclonic or absence) of-ten happen when rapid bilateral diffusion occurs [7] and are reported in a minority of patients. However, the seizure semiology appears to be highly variable during the disease history, with a prevalence of non-motor seizures in the first years of life. From 10% to 53% of subjects are at risk of progressing into status epilepticus (SE), frequently as convulsive SE consisting of repetitive focal motor seizures. Rarely, Non-Convulsive Status Epilepticus (NCSE) might occur, characterized by unresponsiveness, eye deviation, oral automatisms, and distal myoclonic jerks and, in some patients, vegetative symptoms, such as tachycardia and/or abnormal breathing, might also be associated.

#### 4.1.3. EEG Features

Interictal EEG is highly variable. It is reported to be normal in approximately half of patients or characterized by slowing of background activity and focal epileptiform and/or slow wave abnormalities in the remaining half [6]. A photo-paroxysmal response occurs in about one third of patients. Seizures recorded on ictal EEG often arise from the temporal regions [56]. In a case series, Trivisano et al. [7] reported an ictal EEG with seizures independently arising from both hemispheres, even during the same cluster, and seizures characterized by asynchronous and independent discharges on both hemispheres. This pattern is often observed in other genetic epilepsies and resembles epilepsy with migrating seizures of infancy. An example of ictal and interictal EEG characteristic of *PCDH19*-related DEE is shown in Figure 2 and Figure 3.

#### 4.1.4. Neurocognitive Profile and Other Disorders

Epilepsy is associated with mild to moderate-severe ID. However, patients with normal psychomotor development have also been reported. Psychiatric comorbidities are the most frequent and include autistic-like disorders, hyperactive and/or attention-deficit and behavioral disturbances. Obsessive traits and a risk of psychotic behavior may also occur [58].

#### 4.1.5. MRI Features

Initial studies reported mostly normal brain MRIs associated with this condition. Lotte et al. noted normal brain MRI in 81% of 58 patients [59]. A focal cortical dysplasia was identified in 4% of patients, with further suspicion of the same diagnosis in an additional 10% of cases. One patient was noted to have hippocampal sclerosis, and another patient had an arachnoid cyst. However, Kurian et al. described five children with a cortical malformation (four of them had FCD I or IIa, and one had sub-ependymal periventricular nodular heterotopia, a disorder of cortical migration rather than cell proliferation). In two of these patients, histopathology confirmed the diagnosis. Seizure control improved in two patients after epilepsy surgery [60].

#### 4.1.6. Treatment

Epilepsy is known to be highly drug resistant especially in the first years of life. These patients would be on monotherapy very rarely. Different combinations of anti-seizure medications (ASMs) have been explored, and none has proven to be definitively superior. Opposite to Dravet syndrome, definite seizure exacerbation from sodium channel blockers, such as lamotrigine (LTG) and carbamazepine (CBZ), has not been proven.

In a multi-center study, the most effective ASMs for patients with PCDH19 mutations were potassium bromide and clobazam [61]. In clinical practice, various combinations of clobazam (CLB), valproic acid (VPA), and stiripentol are also frequently used. Higurashi et al. emphasized the use of low dose midazolam infusion as an effective treatment modality during seizure clusters [62]. Levetiracetam (LEV) is an effective therapy and should be considered early in the management of the highly refractory clusters of seizures [63]. However, the effect might be transient, with potential worsening during dose reduction. Among third generation ASMs, perampanel (PER) has been reported as efficacious [64]. Intravenous phenytoin (PHT) and phenobarbital (PB) were also used successfully to terminate seizure clusters. Corticosteroids have also been used, particularly during seizure clusters. However, though corticosteroids may suppress seizure clusters, rapid recurrence occurs with no definite long-term benefit in seizures control [8]. A randomized, double-blind, placebo-controlled, phase 2 trial with a synthetic analog of endogenous allopregnanolone, namely ganaxolone, was conducted (VIOLET; NCT03865732). Nevertheless, although ganaxolone was well tolerated and led to a greater reduction in the frequency of clustering seizures compared to placebo, the difference did not reach statistical significance. Finally, ketogenic diet (KD) is reported as an effective adjuvant therapy [65,66,67].

#### 4.1.7. Clinical Outcome

The severity of epilepsy is also extremely variable, from drug-resistant and progressive forms to self-limiting ones. Three clinical stages were identified during *PCDH19* clustering epilepsy: (I) seizure clusters without fever in the first 2 years of life in healthy girls; (II) seizure clusters during febrile illness between 2 and 10 years; (III) after puberty, most of the patients have a spontaneous reduction of seizure frequency, and the most disabling features are ID and behavioral disturbances [8].

### 4.2. CDKL5

Mutations in the X-linked cyclin-dependent kinase like 5 (*CDKL5*, OMIM 300203) gene (previously known as *STK9*, serine/threonine kinase 9) are responsible for a severe encephalopathy with X-linked infantile spasms (ISSX, OMIM 308350) [68,69,70,71].

#### 4.2.1. Genetic Features and Inheritance

To date, various *CDKL5* variants have been identified. Although there are limited data on genotype–phenotype correlation, mutations affecting the catalytic activity of *CDKL5* have been associated with a more severe phenotype [72,73] and missense variants have been correlated with a slightly less severe phenotype compared to truncating variants [74]. Furthermore, a common pathway between *CDKL5* and *MECP2* has been identified, since *CDKL5* phosphorylates the product of the *MECP2* gene [75].

#### 4.2.2. Epileptic Phenotypes

The core features of *CDKL5*-related phenotype are an early onset epileptic encephalopathy, severe developmental delay, deceleration of head growth, impaired communication, hand stereotypies and pronounced hypotonia [9,12]. Males are at the more severe end of the phenotypic spectrum with virtually no motor acquisition [10,76,77]. Epilepsy typically manifests with IS and tonic seizures, starting between the first days and fourth month of life (median 6 weeks) [9,72,78]. A peculiar seizure pattern has been described: “prolonged” generalized tonic–clonic events, lasting 2–4 min, consisting of a tonic-vibratory contraction, followed by a clonic phase with series of spasms, gradually translating into repetitive distal myoclonic jerks [10]. However, other seizure types can occur, including autonomic seizures, myoclonic, atonic seizures, and absences [9]. With regard to EEG features, Buoni and colleagues [79] studied the electro-clinical pattern of epilepsy in three *CDKL5* patients. They emphasized a specific epileptic pattern that consisted of myoclonic epilepsy with refractory seizures and a “unique electroencephalogram (EEG) pattern”. This included diffuse and high-amplitude continuous sharp waves with multifocal spikes, and interictal quasi-periodic diffuse poly-spike or wave discharges. The ictal EEG during tonic seizures is usually characterized by a bilateral and synchronous initial flattening, followed by fast activity discharge and repetitive spike and wave complexes in frontal or central regions. Early EEG findings vary from normal background to moderate slowing, with superimposed focal or multifocal interictal discharges and, in some cases, a burst-suppression pattern [10]. Atypical hypsarrhythmia is often seen in early infancy, while in older patients the interictal EEG usually shows diffuse and high-amplitude sharp waves with pseudo-periodic multifocal spikes, poly-spikes or spike-wave discharges [72] (Figure 4).

#### 4.2.3. Neurocognitive Profile and Other Disorders

Achievement of developmental milestones is severely impaired, with most patients attaining them at a late age. No clear relationship between variant type and milestone attainment was present, although females with a late truncating mutation attained the most milestones. Functional abilities, particularly motor abilities, were related to seizure frequency and refractory to ASMs [79,80]. Neuroimaging has not been systematically analyzed. Several case reports found normal brain anatomy and, rarely, white matter hyper-intensity and cortical atrophy [72].

#### 4.2.4. Treatment

Treatment with ASMs is similar to other drug-resistant epileptic encephalopathies. Cannabidiol based products have attracted significant attention over recent years [81,82]. Fenfluramine has also been observed to be effective for tonic–clonic and tonic seizures [83]. Non-pharmacologic strategies, such as KD and vagus nerve stimulation (VNS), were also proposed [84,85]. Recently, a phase 3 randomized, double-blind, placebo-control trial showed that ganaxolone was effective in reducing seizure frequency [11].

#### 4.2.5. Clinical Outcome

In *CDKL5*-DEE, seizures often remain drug-resistant and developmental milestones are achieved in less than one quarter of cases [9]. A three-step epilepsy phenotype associated with *CDKL5* variants has been described [72], consisting of early epilepsy (stage I), followed by epileptic encephalopathy (stage II), and finally late multifocal and myoclonic epilepsy (stage III). These stages are interspersed with “honeymoon periods” (defined as a seizure-free period longer than 2 months), with a median duration of 6 months [72]. A longer duration of such honeymoon periods has been correlated with better epilepsy outcome, while the developmental impairment proceeds independently [72,86].

### 4.3. MECP2

Mutations in Methyl-CpG-Binding protein 2 (*MECP2*), located on Xq28 and encoding a methyl CpG binding protein, are related to variable clinical phenotypes, based on gender.

In female patients, variants of the *MECP2* gene cause a spectrum of phenotypes ranging from severe encephalopathy to asymptomatic carriers, whose carrier status is detected after investigation for familial Rett syndrome (RTT) [12,87]. The asymptomatic carrier status is favored by extreme X-inactivation skewing, with the mutated allele being largely inactivated [12]. The severe phenotypes include boys presenting with severe early progressive encephalopathy [88], early onset myoclonic epilepsy [89], and early death. The type and location of pathogenic variants dramatically affect the male phenotype, with severe inactivating variants causing neonatal encephalopathy [87] and hypo-morphic alleles yielding to X-linked intellectual disability with or without psychosis [90,91]. *MECP2* has also been implicated in Angelman-like (AS-like) phenotypes [92]. Epilepsy is also frequently described in the *MECP2* Duplication Syndrome, both in males and females. Below, we review epilepsy features in Rett syndrome and in other MECP2-related disorders.

#### 4.3.1. Rett Syndrome (RTT)

Typical Rett syndrome remains by far the most frequent phenotype associated with *MECP2* pathogenic variants. It occurs in approximately 1 in 10,000 females and is the second most common cause of severe ID in females. The currently used diagnostic criteria for RTT include an early neurologic regression, occurring after an initially normal development, that severely affects motor, cognitive, and communication skills. Particularly, one of the main features of RTT is the loss of spoken language and hand use, with the development of distinctive hand stereotypies. After the period of regression, a stage of stabilization and potentially even improvement ensues [93]. Motor impairment constitutes one of the core features of RTT, including the development of gait abnormalities, stereotypic hand movement and the progressive deterioration of motor abilities. Impairment of GABAergic inhibitory circuits within the primary motor cortex has been observed in RTT patients through transcranial magnetic stimulation (TMS) protocols and it has been hypothesized as one of the underlying mechanisms of motor dysfunction [94]. Moreover, in RTT mouse models, significant structural and functional impairments affecting the thalamo–cortical circuitry and the developing somatosensory cortex have been observed. These anomalies have been hypothesized as contributing to the sensorimotor stereotypies characteristic of RTT [95].

In addition to typical RTT, “variant” or “atypical” RTT forms have been recognized. In such atypical RTT forms, many of the clinical features of RTT are present, but the diagnostic criteria are not entirely met and pathogenic variants in *MECP2* are identified in about 50–70% of cases. These variant forms include the early-seizure onset variant, that has been linked to pathogenic variants in the X-linked *CDKL5* gene [72], and the congenital variant, that is distinguished from typical RTT, since development is usually impaired from birth and has been related to *FOXG1* pathogenic variants [96]. Although epilepsy is not the core feature of the typical syndrome, it represents a significant clinical problem in RTT and negatively impacts the quality of life of parents and caregivers of patients with RTT [13]. Epilepsy is frequent and affects approximately 60–80% of patients. Seizures usually appear during the regression stage, at a median age of 3–4 years, and may play an encephalopathic effect [12,97,98,99,100]. Early age of onset predicts a more severe course of seizures and represents a risk factor for Electrical Status Epilepticus During Sleep (ESES) occurrence. The *BDNF* val/met polymorphism has been correlated with earlier onset of seizures, whereas some *MECP2* variants have been correlated to increased risk for epilepsy and/or epilepsy occurrence (i.e., p.R133C, p.R255X, p.T158M, p.C306C) [98,99]. There is no specific semiology for seizures in RTT; all seizure type have been reported, particularly focal seizures, followed by tonic–clonic, tonic and myoclonic seizures. Cortical reflex myoclonus, manifested as multifocal, arrhythmic, and asynchronous jerks involving distal limbs, is frequently observed in RTT and exhibits neuro-physiological features, including a prolonged intra-cortical delay of the long-loop reflex [12,101]. Further differentiation between seizures and paroxysmal non-epileptic events is critical and not always simple. Long-term video-EEG monitoring is a safe diagnostic tool that provides a high diagnostic yield. It is essential for characterizing paroxysmal events, differentiating breathing and autonomic dysfunction and motor phenomena, and evaluating the treatment response to ASMs. The EEG patterns are neither diagnostic nor patho-gnomonic of RTT and include progressive generalized background slowing and/or loss of the occipital dominant rhythm, with further theta and delta slowing as these children continue to regress developmentally [92,102] (Figure 5).

Due to limited experience of epilepsy treatment, there are no definitive recommendations regarding treatment approach with the ASMs currently available. CBZ and VPA were the most widely used. However, CBZ has been reported to aggravate ESES pattern [103]. Alternative therapies, such as VNS, KD or cannabinoids and neuro-steroids, are also reported in literature. The relationship between epilepsy and global severity is not clear. In general, individuals with seizures are considered to have greater overall clinical severity, particularly greater impairment of ambulation, hand use, and communication [12,13].

To conclude, additional features may be observed in RTT individuals, partly included within the supportive criteria, such as impaired nociception, sleep disturbances, and hypersalivation due to oral motor dysfunction [104,105].

#### 4.3.2. Other MECP2-Related Disorders

As an X-linked disorder, it had been suspected that mutations in *MECP2* led to embryonic lethality in males. However, males harboring *MECP2* pathogenic variants have been reported, defining a new entity: Male RTT Encephalopathy” [106], which includes the key RTT diagnostic feature, developmental regression, but also a more severe clinical course than typical RTT in females. A genotype–phenotype study in boys with *MECP2* mutations showed a range of severity from fatal neonatal encephalopathy to psychiatric abnormalities (i.e., schizophrenia or bipolar disorder), depending on the type and position of the mutation [107]. Males harboring *MECP2* variants fall into four categories:Severe neonatal encephalopathy and infantile death. *MECP2* variant is usually passed on by a mildly symptomatic or asymptomatic mother. Clinically the scenarios can be spontaneous miscarriages or, if born, neonatal encephalopathy, respiratory arrest and seizures; death occurs within 2 years.Classical male RTT patients have at least partial Klinefelter’s syndrome (XXY karyotype) or other somatic mosaicism and the symptoms are similar to female RTT patients [108,109].Less severe neuro-psychiatric symptoms. In these cases, the MECP2 variants are less severe than those in female RTT patients, and symptoms are heterogeneous and overlap with features of Angelman syndrome (intellectual disability and motor abnormalities).MECP2 duplication syndrome, with gain in MECP2 dosage. The clinical phenotype is characterized by hypotonia, severe ID, recurrent lung infections, absent or limited speech and walking, seizures, motor spasticity and muscle stiffness, and 50% of affected individuals usually die before the age of 25 years [110,111].

Many males harboring RTT-causing MECP2 variants present with epilepsy, which is often severe and medically refractory. Other symptoms, such as regression, developmental delay, absence of language, and motor deficits, may also occur.

#### 4.3.3. The MECP2 Duplication Syndrome (MDS)

MECP2 duplication syndrome (MDS) is a severe, clinically recognizable X-linked recessive neurodevelopmental syndrome caused by gain-of-function duplications of *MECP2* on the long arm of the X chromosome (Xq28). The MDS phenotype shares some features with RTT. It is characterized by neonatal or infantile hypotonia, severe to profound ID with poor speech development, lack of walking acquisition or motor delay with ataxic gait and progressive spasticity, stereotypies, recurrent respiratory infections, gastrointestinal problems, and mild facial dysmorphisms. Developmental regression is sometimes observed [12,14,106]. The main differences with RTT are that children with MDS are much more likely to be male and have recurrent pulmonary infections [106]. Epilepsy has been reported in over approximately 60% of children with MDS [14,106,112], usually with onset later than in RTT, with a median age of 8 years. Seizures tend to be refractory to medications [107,113].

Seizures have been described as focal or generalized, with heterogeneous semiology, including atypical absences, generalized convulsive, myoclonic, and focal seizures, with frequent status epilepticus [112]. However, atonic seizures have been reported as a frequent manifestation, with or without a myoclonic component [15]. Approximately half of MDS individuals meet the criteria for Lennox–Gastaut syndrome [107]. The most frequent EEG pattern was the occurrence of generalized slow spike and wave asynchronous discharges with fronto-temporal predominance. Sleep electroencephalography studies also demonstrated abnormal background activity; spindles and K complex were often abnormal in morphology and amplitude (Figure 6).

The majority of *MECP2* duplications are inherited, while de novo duplications are rare [114]. Most cases are inherited maternally; paternal inheritance is exceedingly rare, and there have been only four reported cases of paternal inheritance, in which the male patients inherited the gene duplication from an unbalanced X/Y translocation [115]. As an X-linked disorder, males are predominantly affected, while penetrance in female patients is variable due to skewed X-inactivation [116].

Females with one copy of *MECP2* duplication are usually asymptomatic carriers, since XCI is often skewed and preferentially inactivates the duplication-bearing X chromosome. However, female patients with *MECP2* duplication have been reported [117]. The molecular genetic pathogenesis of symptomatic female patients included (1) unbalanced translocation between X chromosome and autosome, (2) no skewing of X-chromosome inactivation and (3) skewed XCI where the normal chromosome is preferentially inactivated. The clinical spectrum in rare symptomatic female includes non-specific mild to moderate ID to severe phenotype with drug-resistant epilepsy similar to affected males [118]. Although genotype–phenotype correlations in MDS are limited, larger duplication size has been correlated to increased severity. Additionally, duplication of neighboring genes could contribute to additional clinical phenotypes. For example, the Filamin A gene (*FLNA*) duplication has been proposed to contribute to intestinal and bladder dysfunction and development of distinct facial features amongst individuals with MDS [119,120].

## 5. X-Linked Neuronal Migration Disorders

Neuronal Migration Disorders (NMDs) are a group of various malformations of cortical development related to dysregulated neuronal differentiation and migration, including lissencephaly, subcortical band heterotopia (SBH) and periventricular nodular heterotopia (PNH) [121]. Many X-linked NMDs occur because of genetic mutations which are often familial. In female probands, the severity of the phenotype is usually attenuated by X-inactivation. Moreover, skewed X-inactivation may occur and lead to further heterogeneity of phenotype in female probands. Therefore, females with mild symptoms may pass on X-linked NMD mutations to male offspring who then present with a more severe form of the disorder [19].

### 5.1. ARX

The Xp21.3 gene *ARX* encodes a transcription factor involved in tangential migration and differentiation of GABAergic interneurons in the cortex. Pathogenic variants in *ARX* lead to functional alterations affecting microtubule network and RNA metabolism [122]. A spectrum of phenotypes is related to *ARX* pathogenic variants, which range from non-syndromic X-linked ID (XLID), developmental and epileptic encephalopathies (e.g., EEIE, IESS), Partington and Proud syndromes to X-linked lissencephaly with abnormal genitalia (XLAG) [16,17,18,19]. The *ARX*-associated disorders may be divided into two categories based on presence or absence of brain malformations. NMDs are present in XLAG, Proud syndrome, and hydranencephaly with abnormal genitalia [1]. Particularly, XLAG is characterized by corpus callosum agenesis, neonatal refractory epilepsy, hormone dysregulation, temperature instability and lissencephaly, which is more severe caudally [123]. Males are primarily affected by XLAG, while females usually show milder phenotype [124].

To date, various *ARX* variants have been identified, including a 24-bp duplication in Exon 2 (c.428_451dup), expansion in the first or second poly-alanine tract, frameshift, and deletion and point mutation both within and outside of the homeodomain [125,126]. Moreover, some genotype–phenotype correlations have been highlighted. Truncation mutations and missense mutations in the *ARX* homeodomain result in severe brain malformation phenotypes, while missense mutations outside of the homeodomain and expansion of the poly-alanine tracts are associated with non-malformation phenotypes [125,126,127]. In the context of developmental malformations, *ARX* related syndromes can be thus considered as “inter-neuronopathies” and the epilepsy and ID would be directly related to the reduction of inhibition [128].

### 5.2. DCX

The doublecortin (*DCX*) gene, mapped to Xq23, encodes for a microtubule-associated protein which interacts with lissencephaly 1 protein (*LIS1*), expressed at high level in the orbitofrontal region [1,19]. In hemizygous males, *DCX* pathogenic variants are usually responsible for lissencephaly and present with agyria or pachygyria with global developmental delay, severe ID, and seizures. Heterozygous females present with a milder phenotype, showing mixed pachygyria and SBH or SBH alone, presumably due to random X-inactivation [129]. The epileptic phenotype is heterogeneous: most reports suggest an early onset of seizures, during the first year of life in most cases; however, the age at onset varied between the first month of life and 17 years [20]. SBH has been observed in a minority of males harboring *DCX* variants affecting non-coding gene regions or somatic mosaicism of the *DCX* mutation, expressed only in a subset of neurons [130]. A wide range of *DCX* mutation types have been identified in lissencephaly and SBH, including missense, nonsense, frameshift with insertion or deletion, splice site and deletion of exons [20]. Frameshift variants were observed to be more commonly de novo, producing more severe phenotypes, while missense mutations inferred milder developmental and cognitive defects [131].

### 5.3. FLNA

The filamin-A (*FLNA*) gene, mapped to Xq28, is the most common genetic cause of X-linked bilateral periventricular nodular heterotopia (BPNH), that usually shows anterior predominance [22,132]. Other genes associated to BPNH in a minority of cases are the Fragile X mental retardation 1 (*FMR1*) and L1 Cell Adhesion Molecule (*L1CAM*) genes [133]. *FLNA* encodes a structural protein involved in connecting membrane proteins to the cytoskeleton. Particularly, in the developing brain, it is involved in radial migration of excitatory neurons [23,134]. *FLNA* is responsible for 49% of all bilateral PNH cases, and for the 77% of female cases. In familial cases, usually heterozygous female individuals, *FLNA* pathogenic variants are truncating and cause a complete loss of function. Conversely, in male patients, pathogenic variants are usually de novo and missense, therefore they cause a milder protein dysfunction [19,22]. Systemic manifestations may be present, including facial dysmorphism, cardiac abnormalities, excessive laxity of skin and joints, and intestinal dysfunction. The electro-clinical pattern may be variable and specific features have not been reported. However, in approximately a third of patients, seizures are resistant to multiple ASMs [135].

### 5.4. X-Linked Polymicrogyria (PMG) Genes

Polymicrogyria (PMG) is a malformation secondary to abnormal post-migrational development, which is heterogeneous in cause and has been reported in association with various genetic disorders [121]. X-linked PMG is usually bilateral and symmetric, particularly bilateral peri-sylvian PMG and bilateral frontal PMG. Epilepsy is observed in approximately 80% of cases. Among identified X-linked PMG-associated genes, the DEAD-Box Helicase 3 X-Linked (*DDX3X*) gene is associated with the most PMG cases together with Sushi Repeat Containing Protein X-linked 2 (*SRPX2*). Other genes identified in fewer cases are *NSDHL*, *CUL4B* and *OFD1*. To date, all PMG cases with *DDX3X* mutations are females with heterozygous de novo variants, while in males *DDX3X* variants seem to be not compatible with life. Conversely, *SRPX2* and *NSDHL* have been observed to affect males, owing to skewed X-inactivation in females [19]. The *SRPX2* gene was initially related to Rolandic seizures, ID, speech delay, oro-facial dyspraxia, and abnormalities in brain speech areas [136].

## 6. Other Emerging X-Linked DEE Genes

### 6.1. SLC9A6

The *SLC9A6* gene, mapped to Xq26.3, encodes a sodium/hydrogen exchanger (NHE6) localized both on plasma membrane and mitochondrial inner membrane [137]. The NHE6 is involved in many cellular processes critical to normal nervous system development and responsiveness to noxious stimuli [138]. Loss of function variants in this gene have been associated with the Christianson syndrome (OMIM 300243) [139]. This syndrome usually presents with severe ID or regression, early-onset seizures, hypotonia, microcephaly, impaired ocular movements, ataxia and facial dysmorphisms in males. A milder phenotype with learning difficulties, behavior disorder and mild-moderate ID is observed in female carriers. Epilepsy, most often drug-resistant, usually manifests with generalized tonic-clonic seizures, atypical absences, focal-onset seizures with impaired awareness. The EEG pattern in some cases shows slow rhythmic activity of high amplitude, which resembles the pattern in Angelman syndrome [24,25].

### 6.2. SLC35A2

The *SLC35A2* gene, mapped to Xp11.23, encodes a UDP-galactose transporter, required for protein and lipid glycosylation in the Golgi apparatus. Pathogenic variants in this gene are responsible for a congenital disorder of glycosylation (CDG) in females, that comprises early-onset epileptic encephalopathy, ID, hypotonia, facial dysmorphisms, skeletal abnormalities, microcephaly in some cases, abnormal brain imaging. Among these latter, white matter abnormalities and cerebellar atrophy are the most frequent.

The epileptic phenotype is often consistent with infantile epileptic spasms syndrome, with IS beginning within the first year of life, and a hypsarrythmic EEG pattern. Females are inherently mosaic for X-linked genes owing to X-chromosome skewed inactivation. Somatic variants in *SLC35A2* have been identified in some cases, both males and females with focal cortical dysplasia [26,27,28].

### 6.3. SYN1

The synapsin-1 (*SYN1*) gene, mapped to Xp11.23, encodes the synaptic vesicle protein synapsin I, which is involved in synaptogenesis and regulation of neurotransmitters release. Mutations in this gene are considered responsible for epilepsy, learning difficulties, specific language impairment and mild autistic spectrum disorder (ASD), in male carriers. Particularly, *SYN1*-associated epilepsy is characterized by reflex seizures triggered by bathing or showering. Ictal EEG recordings and ictal single-photon emission computed tomography (SPECT) have shown that the epileptic focus is localized in the temporo-insular region. Female carriers usually present with a milder phenotype, with reading impairments and febrile seizures but no chronic epilepsy [29,30,31].

### 6.4. ARHGEF9

*ARHGEF9* pathogenic variants have been associated with early onset developmental and epileptic encephalopathy. Epilepsy frequently manifests with tonic-clonic seizures. However, seizure semiology is variable with many patients showing multiple seizure types. Seizures are often drug resistant. Brain MRI may not show anomalies, although polymicrogyria, hippocampal sclerosis and delayed myelination have been described in some cases. Female cases have been reported, usually showing strongly skewed X-inactivation in favor of the abnormal X-chromosome. *ARHGEF9* encodes the protein collybistin, a brain-specific GDP/GTP-exchange factor which is involved in inhibitory synaptic transmission, specifically in postsynaptic glycine and gamma-aminobutyric acid receptor clustering via its interaction with the inhibitory receptor anchoring protein gephyrin [32,33,34].

### 6.5. ATP6AP2

*ATP6AP2,* mapped to Xp11.4, encodes an accessory unit of vacuolar ATPase, an essential lysosomal enzyme. This gene is responsible for XLID with early epilepsy onset. Seizures usually begin between four and 14 months of age, and consist of primarily generalized tonic-clonic, brief atonic and myoclonic seizures. The ictal EEG shows generalized poly-spike and waves discharges before seizures. Scoliosis, ataxia and hyporeflexia may occur. Parkinsonism has been reported in a minority of cases. Brain MRI may show cerebellar atrophy and/or thinning of the corpus callosum, although it has been described as normal in the majority [35,36,37].

### 6.6. IQSEC2

The *IQSEC2* gene, mapped to Xp11.22, encodes the IQ motif and Sec7 domain-containing protein 2, that is highly expressed in the brain and is clustered at excitatory synapses [140]. The phenotypic spectrum of *IQSEC2*-associated disorder includes X-linked epileptic encephalopathy and non-syndromic XLID with epilepsy. Particularly, de novo pathogenic variants in this gene have been observed in both males and females presenting with epileptic encephalopathy. In these cases, seizures usually manifest between 8 months and four years of age; seizures are of various type (i.e., atonic, myoclonic, epileptic spasms, atypical absences, generalized tonic–clonic seizures) and resistant to multiple ASMs. In some *IQSEC2*-related DEE patients a Lennox-Gastaut-like phenotype and a Rett-like phenotype in females have been described. However, in the majority seizure type and EEG pattern are variable and do not meet the criteria for a specific epileptic syndrome. Additional clinical features include autistic features, hypotonia, strabismus, microcephaly, craniofacial dysmorphisms. Brain MRI may show abnormal findings, including white matter changes, thin corpus callosum and cortical atrophy. Conversely, patients harboring inherited *IQSEC2* pathogenic variants usually present with moderate to profound ID and generalized epilepsy, sometimes responsive to ASMs [38,39].

### 6.7. NEXMIF

The neurite extension and migration factor (*NEXMIF*) gene, previously called *KIAA2022*, is an X-linked gene involved in early brain development. *NEXMIF* loss of function variants have been reported in individuals with ID, ASD, and epilepsy. In females, pathogenic variants in this gene are responsible for a DEE overlapping with epilepsy with eyelid myoclonia (EEM) and epilepsy with myoclonic–atonic seizures (EMAtS). Seizures are predominantly generalized, including absences with eyelid myoclonia, myoclonic seizures, myoclonic–atonic or atonic seizures. The EEG predominant findings are generalized spike wave or poly-spike wave discharges. Photosensitivity and/or eye closure sensitivity have been observed as either a clinical or electro-encephalographic phenomenon in about 20–30% of patients with epilepsy. Males are more severely affected than females, showing more severe ID. However, seizures are less common and, when present, usually consist of epileptic spasms. Additional clinical features include infantile hypotonia, ataxia, microcephaly, gastro-esophageal reflux disease, strabismus, and dysmorphic features. Brain MRI does not usually show major anomalies [40,41]. Advanced neuroimaging techniques have revealed brain structural and functional changes, especially in the visual pericalcarine cortex and in the middle frontal gyrus, which are involved in several visuo-motor function and therefore in epilepsy phenotypes with visual sensitivity and eyelid myoclonia [141].

### 6.8. PIGA

*PIGA* encodes an enzyme involved in the GPI anchor biosynthesis pathway, which is important for anchoring proteins to the cell membrane. GPI-anchored proteins are thought to be involved in neurogenesis, complement regulation, and embryogenesis [142]. The phenotypic spectrum of PIGA-related congenital disorder of glycosylation (*PIGA*-CDG) spans from early onset epileptic encephalopathy to multiple congenital anomaly syndrome. Most individuals present with hypotonia, moderate to profound developmental delay and ID, and intractable seizures. Congenital heart anomalies are common. Epilepsy is present in almost all patients, usually with onset within the first year of life. Myoclonic and/or tonic seizures often with apnea, sometimes evolving to bilateral tonic clonic seizures, are common. Focal seizures, migrating focal seizures, atonic seizures and gelastic seizures have also been reported. Epileptic spasms are usually seen in patients with severe or profound developmental delay. Fever-sensitivity may be common. The EEG pattern typically shows diffuse background slowing and high amplitude multifocal or diffuse spike/sharp and slow waves. Brain anomalies usually include cortical and subcortical volume loss, cerebellar and/or brainstem hypoplasia, white matter immaturity, and corpus callosum dysgenesis; focal cortical dysplasia and hippocampal sclerosis have been observed in some cases. Seizures are often resistant to multiple ASMs and sudden unexpected death in epilepsy (SUDEP) may occur. However, seizure freedom achievement has been observed in few cases when adding KD or pyridoxine [42,43,44].

### 6.9. ALG13

Deficiency of asparagine-linked glycosylation 13 (*ALG13*) causes a congenital disorder of glycosylation (CDG), usually with normal transferrin electrophoresis and consisting of a DEE. ALG13-related DEE mainly affects females carrying the recurrent *ALG13* de novo variant p.Asn107Ser. It usually manifests as IESS, with IS beginning within 6 months of life and an EEG hypsarrythmic pattern. Severe to profound developmental delay with regression is present, as well as hypotonia, movement disorders, and sometimes dysmorphic features. Epileptic spasms persist beyond 2 years of age in 60% of individuals. After the age of 2 years, a distinctive EEG pattern comprising paroxysms of fast activity with electro-decremental response lasting a few seconds without clinical correlate has been observed in some patients. Tonic seizures are also common and evolution to Lennox Gastaut syndrome may occur. Brain MRI usually shows non-specific anomalies, including white matter volume loss, corpus callosum dysgenesis and cortical atrophy [45,46].

### 6.10. FHF2/FGF13

Fibroblast growth factor homologous factors (FHFs) are intra-cellular proteins which regulate voltage-gated sodium (Nav) channels in the brain and other tissues.

FHF2/FGF13, mapped to Xq26.3-q27.1, is highly expressed in the developing brain and hemi- and heterozygous gain of function missense variants in the N-terminal of the A isoform of this gene cause infantile-onset DEE. Seizures usually have focal onset with motor features, apneas, and oro-alimentary automatisms. Epileptic spasms, absences, atonic and generalized tonic–clonic seizures have also been observed. The EEG has shown a temporal focus in some cases. Seizures tend to resist multiple ASMs. Moreover, in one individual, seizures resisted left anterior temporal lobectomy and partial amygdalohippocampectomy [47,48]. A maternally transmitted balanced translocation between chromosome X and 14 involving *FGF13* has been identified in a family with Genetic Epilepsy with Febrile Seizures Plus (GEFS+) and temporal lobe epilepsy [143].

### 6.11. GRIA3

The X-linked *GRIA3* gene encodes for the subunit 3 of the glutamate ionotropic AMPA receptor. Pathogenic variants in this gene mainly affect males and are associated with ID, dysmorphic features, and epilepsy. Epilepsy onset is usually early and myoclonic seizures are frequent. NCSE with subtle myoclonia lasting 20 days has been reported in one case. However, a gain of function variant in *GRIA*, enhancing synaptic transmission, has been identified in a female affected with DEE [48,49]. No typical electro-physiological or brain MRI findings were reported. Indeed, the latter is often reported as normal. Treatment of seizures remains largely empirical, and individual prescribing based on the mechanism of action is generally not possible. However, a trial with perampanel (PER), the only available drug active on AMPA receptor, is conceivable [49,50].

### 6.12. SMC1A

Pathogenic variants in *SMC1A* are often dominant-negative and cause Cornelia de Lange syndrome (CdLS). This syndrome is characterized by growth retardation, ID, distinct facial features, and major malformations, including heart anomalies. Multiple genes are associated with CdLS, including heterozygous pathogenic variants in *NIPBL*, *RAD21*, *SMC3*, and *BRD4*, or hemizygous pathogenic variants in *HDAC8* and *SMC1A*. Among these, *NIPBL* is the most common gene, associated with 60–70% of CdLS patients. *SMC1A* variants are found in about 5% of CdLS female patients, often associated with mild CdLS phenotypes. However, rare *SMC1A* loss of function variants have been identified in females presenting with a more severe phenotype of early-onset DEE. *SMC1A*-related DEE in these females manifests with a Rett-like phenotype and drug-resistant clustering seizures. A differential dosage effect of *SMC1A* loss of function variants has been closely associated with the manifestation of DEE phenotype [51,52,53].

## 7. Conclusions

In this narrative review, we provide a detailed description of the main electro-clinical features of epilepsies associated with X-linked genes. XLEs are a heterogeneous expanding group of epileptic syndromes and DEEs. They differ from epilepsies caused by pathogenic variants in autosomes because of a greater phenotypic variability, which could hinder their recognition. Different modes of X-linked transmission, different types of pathogenic variants and further modifying factors, including epigenetic regulation and X-chromosome inactivation, underly XLEs’ phenotypic heterogeneity. XLE is well described for some genes, where it represents the main symptom, while it is less characterized for other more recently identified genes. The epileptic phenotype is generally quite severe, with DEE occurring especially in males. The age of onset, albeit variable, is often in infancy and with epileptic spasms. Early onset forms have usually greater risk of drug resistance. ID and neurodevelopmental disorder are constant and, compared to other forms of epilepsy, their progression is usually independent from seizures.

The epilepsy treatment in these conditions is largely based on anti-seizure medication, as tailored treatment is not available to date.

We acknowledge a limitation of this review as it does not cover recurrent CNVs affecting the X chromosome and featuring epilepsy.

Future research will likely lead to the identification of new genes responsible for XLE and to a better characterization of the genotype–phenotype correlations related to known genes. We hope that an updated XLE narrative review will support clinicians in the diagnosis and treatment of patients with epilepsy showing suspected X-linked inheritance.

## Figures and Tables

**Figure 1 ijms-25-04110-f001:**
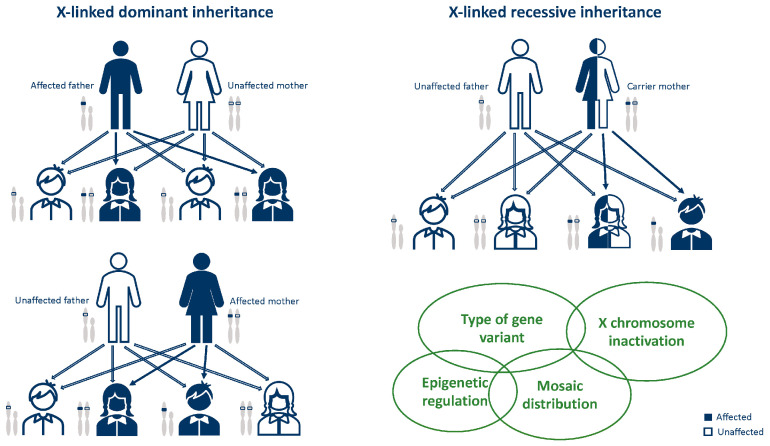
X-linked modes of inheritance, influenced by X-chromosome inactivation in females, epigenetic regulation, mosaic distribution of pathogenic variants.

**Figure 2 ijms-25-04110-f002:**
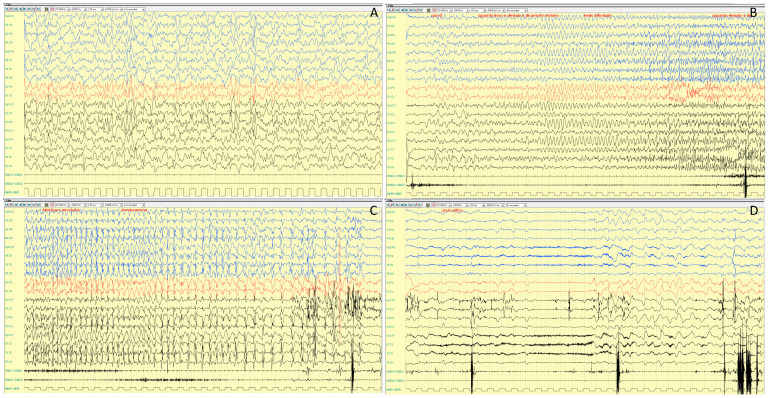
Poly-graphic recording, including EEG and electromyography (EMG) channels of a focal seizure recorded at 7 year and 3 months of age in a girl with *PCDH19*-related DEE. Interictal sleep EEG (**A**) showing slow background activity with poor representation of physiological sleeping figures. Seizure onset (**B**) with a bilateral ictal theta activity over both parieto-temporal regions, followed by bilateral high amplitude slow waves (**C**) and a rhythmic fast ictal discharge persisting over the bilateral temporal regions. Twelve seconds later (**D**), a sudden termination of the ictal discharge with electro-decrementation is visible. Seizure semiology: (1) eyes opening and staring; (2) behavior arrest; (3) right eye deviation, bilateral blinking and minor arm and hand movements, followed by desaturation and post ictal sleep. Poly-graphic recording, including EEG (bipolar montage: blue lines are the right electrodes, black lines are the left electrodes; red lines are the vertex electrodes) and electromyography (EMG1: left deltoid; EMG2: right deltoid) of a focal seizure recorded.

**Figure 3 ijms-25-04110-f003:**
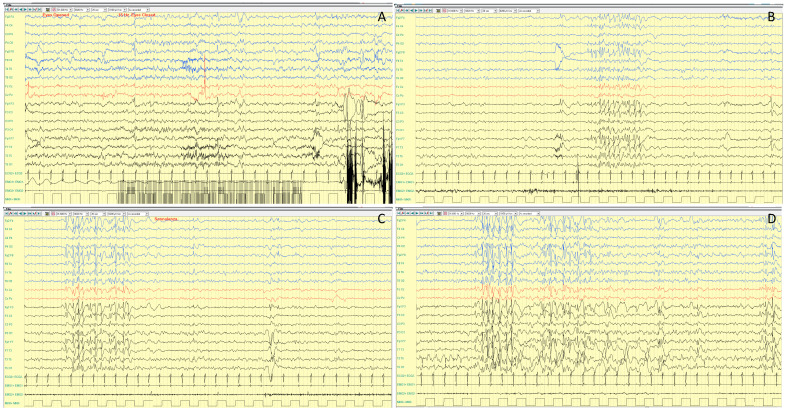
Poly-graphic recording, including EEG and electromyography (EMG) channels, in a 10 year and 2 months years old girl with *PCDH19*-gene related DEE. Interictal awake (**A**,**B**) and sleep (**C**,**D**) EEG study showing slow background activity and bilateral frontal paroxysmal activity, which increases during sleep, spreading over the temporal regions. Poly-graphic recording, including EEG (bipolar montage: blue lines are the right electrodes, black lines are the left electrodes; red lines are the vertex electrodes) and electromyography (EMG1: left deltoid; EMG2: right deltoid) in a 10 year and 2 months.

**Figure 4 ijms-25-04110-f004:**
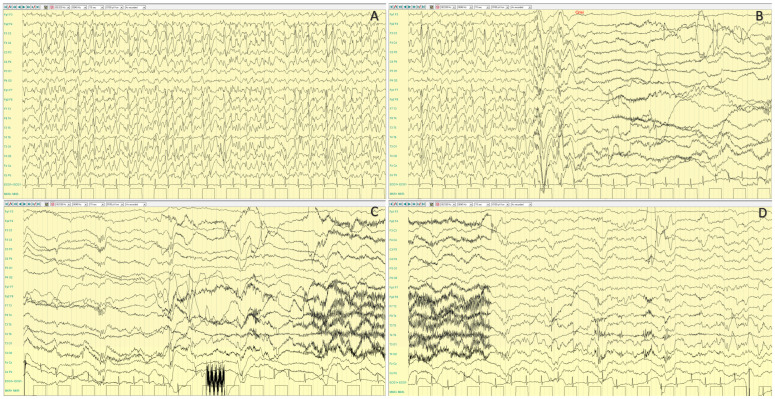
EEG recording (bipolar montage) of a 9 year and 2 months old girl with CDKL5-gene related DEE. (**A**) Interictal EEG during sleep showing continuous spikes and slow wave activity, prominent over the frontal regions. Sleep figures are poorly represented. (**B**) Ictal recording showing a tonic seizure corresponding to an electro-decremental change (**B**) followed by low voltage diffuse fast activity and high-voltage sharp waves (**C**,**D**), lasting 40 s.

**Figure 5 ijms-25-04110-f005:**
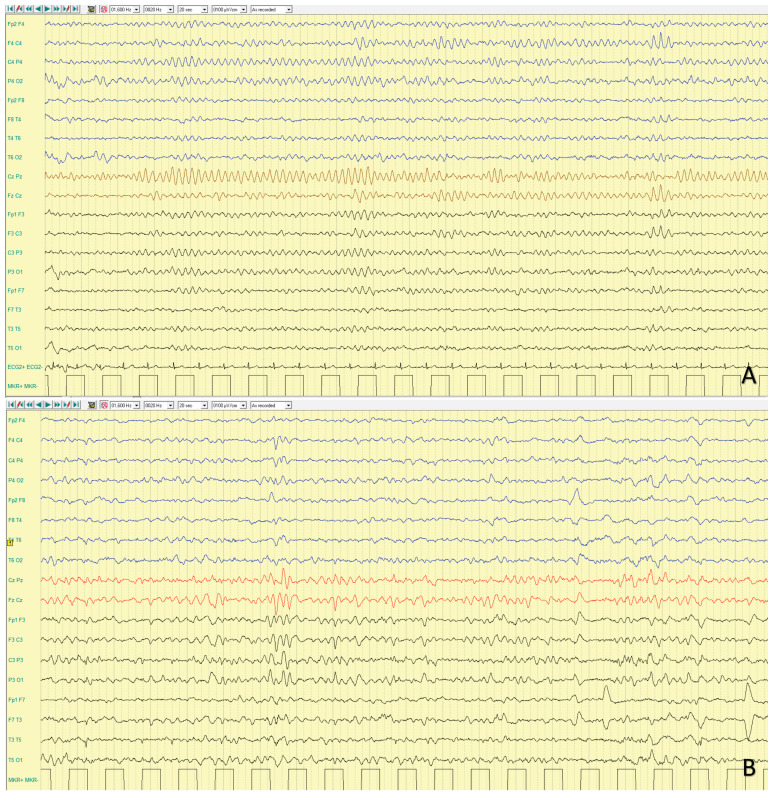
Video-EEG recording of a 14 year and 6 months old girl with typical Rett Syndrome *MECP2*-related. Interictal EEG during wakefulness showing a background activity characterized by monomorphic theta activity (6 Hz), topographically undifferentiated (**A**). During sleep, asymmetrical and disorganized slow activity was present, with poor physiological sleep figures representation (**B**). Video-EEG recording (bipolar montage: blue lines are the right electrodes, black lines are the left electrodes; red lines are the vertex electrodes).

**Figure 6 ijms-25-04110-f006:**
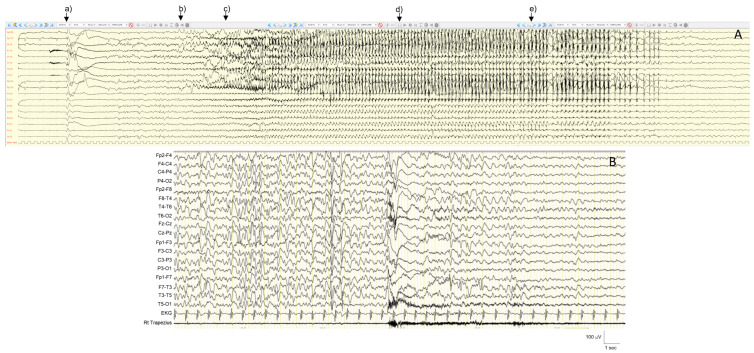
Electroencephalographic (EEG) recording (performed at 25 years old) of a seizure in a female patient with *MECP2* duplication syndrome (MDS) and clinically featuring a Lennox Gastaut Syndrome (**A**). Ictal EEG starts with a generalized wave with an electro-decremental component followed by desynchronization of the background activity; after 6 s, slow delta activity with interspersed spikes appears. After about 20 s, irregular 2–2.5 Hz, high voltage spike-wave or poly-spike–wave complexes appear, with diffuse wide spreading prevalent over the bilateral frontal regions. The event terminates with post ictal delta activity (1.5 Hz) (**A**). Clinically, the episode is characterized by: (a) sudden jerk involving the proximal limbs with impairment of awareness; (b) neck and face myoclonia; (c) right eyes and neck deviation (d) prolonged facial and right distal upper limb myoclonia which subsequently (e) involves asynchronously the left distal upper limb. (**B**) Polygraphic recording including EEG and EMG (Right Trapezius) recording of a male patient affected with MDS and showing slow background activity, diffuse spike and wave epileptiform activity and an ictal spasm of the upper limbs time-locked with the electrodecremental high amplitude wave.

**Table 1 ijms-25-04110-t001:** Electro-clinical features related to the X-linked genes reviewed.

Gene	Locus	Inheritance		Gene–Phenotype Relationships (Incidence If Available)	Main Epileptic Phenotype	Epilepsy Onset (Age)	SE	Main EEG Features		Neuro-Developmental Outcome	MRI Features	Other Features	Specific Treatment	Drug Resistance	References
								Interictal	Ictal						
*PCDH19*	Xq22.1	XL		DEE 9 (1/42,000 live births)	Clusters of FS, focal S: hypomotor and affective S	3–36 mo	10% to 53%	Bilateral SW, rare focal epileptiform abnormalities	Diffuse or frontal and temporal focal discharges (both hemispheres)	ID, ASD, psychiatric disorders. Normal cognitive outcome is also reported	Normal	None	BDZ; Corticosteroids during the clusters; stiripentol	Not constant	Marini, 2012 [6]; Trivisano, 2018 [7]; Samanta, 2019 [8]; Zuberi, 2022 [9]
*CDKL5*	Xp22.13	XLD		DEE 2 (1/40,000-1/60,000 live births)	IS, tonic and myoclonic S, ‘prolonged’ GTCS	3–4 mo	rare	Slow background; SB; multifocal epileptiform activity	Bilateral synchronous initial flattening, followed by repetitive SW and spikes	Severe developmental delay, ID	Normal; progressive brain atrophy	Microcephaly, hypotonia	Ganaxolone	Yes	Zuberi, 2022 [9]; Melani, 2011 [10]; Knight, 2022 [11]
*MECP2*	Xq28	RTT	XL, XLR, XLD	RTT (1 in 10,000 live female births) ASD susceptibility, X-linked 3; Encephalopathy, neonatal severe; IDD, X-linked syndromic 13; IDD, X-linked syndromic, Lubs type	Focal or GTCS	5–10 Y	rare	Unusual fast/theta rhythmic activity	No peculiar pattern	Psycho-motor regression, ID, ASD	Normal	Microcephaly, breathing abnormalities, hand stereotypies, myoclonus, mouthing, hypotonia	NR	30%	Guerrini, 2012 [12]; Glaze, 2010 [13]
		MECP2-dup			Focal or GTCS; atonic seizures or drop attacks	9–13 Y	rare	Unusual fast/theta rhythmic activity, multifocal spike discharges, generalized SW activity, “extreme spindles” during the sleep	Unusual fast rhythms and erratic myoclonic jerks evolving in spike and wave discharges with anterior predominance	ID, ASD		Mild facial dysmorphisms, recurrent infections, dyskinetic movements, hypotonia, evolving to spasticity	NR	Yes	Guerrini, 2012 [12]; Ramocki, 2010 [14]; Vignoli, 2012 [15]
*ARX*	Xp21.3	XLR		DEE 1; Hydranencephaly with abnormal genitalia; IDD, X-linked; Lissencephaly, X-linked 2; Partington syndrome; Proud syndrome	Polymorphic: IS, myoclonic S	First mo	NR	Hypsarrhythmia, multifocal, asynchronous, epileptiform activity	NR	PMD, ID	Corpus callosum agenesis, lissencephaly	Microcephaly, hypotonia, abnormal genitalia and Partington syndrome	NR	Yes	Kitamura, 2002 [16]; Kato, 2004 [17]; Marsh, 2012 [18]; Edey, 2023 [19]
*DCX*	Xq23	XL		Lissencephaly, X-linked; Subcortical laminal heterotopia, X-linked	IS	first Y	NR	NR		PMD, ID	Lis-sencephaly, SBH	NR	NR	Yes	Bahi-Buisson, 2013 [20]; Chou A, 2009 [21]
*FLNA*	Xq28	XL, XLR, XLD		Cardiac valvular dysplasia, X-linked; Congenital short bowel syndrome; Fronto-metaphyseal dysplasia 1; Heterotopia, periventricular, 1; Intestinal pseudo-obstruction, neuronal; Melnick-Needles syndrome	Focal S	First mo-adult	NR	Focal slowing-epileptic discharges or normal	NR	Normal to ID	BPNH	Dysmorphic features, cardiac disease, skin and joints abnormalities		Mild to DRE	Parrini, 2006 [22]; Cannaerts, 2018 [23]
*SLC9A6*	Xq26.3	XL		IDD, X-linked syndromic, Christianson type	Focal or GTCS	NR		slow rhythmic activity of high amplitude	NR	ID or regression, behavior disorder	Asymmetric atrophy most prominent in left frontal and parietal cortex	Hypotonia, microcephaly, impaired ocular movements, ataxia and facial dysmorphisms	NR	Yes	Schroer, 2010 [24], Sinajon, 2016 [25]
*SLC35A2*	Xp11.23	SMo, XLD		Congenital disorder of glycosylation, type IIm	IS	First Y	NR	hypsarrythmic	NR	ID	white matter abnormalities and cerebellar atrophy; FCD	hypotonia, facial dysmorphisms, skeletal abnormalities, microcephaly	NR		Kodera, 2013 [26]; Winawer, 2018 [27]; Ng, 2019 [28]
*SYN1*	Xp11.23	XL		Epilepsy, X-linked 1, with variable learning disabilities and behavior disorders; IDD, X-linked 50	Reflex S, FS	NR			Focus over temporo-insular regions.	Learning difficulties, specific language impairment and mild ASD	NR				Garcia, 2004 [29], Giovedi, 2004 [30] Nguyen, 2015 [31]
*ARHGEF9*	Xq11.1	XL		DEE 8	GTCS	NR					Poly-microgyria, hippocampal sclerosis and delayed myelination	NR		Yes	Shimojima, 2011 [32], Alber, 2017 [33]; Yang, 2022 [34]
*ATP6AP2*	Xp11.4	XLR		Parkinsonism with spasticity, X-linked; Congenital disorder of glycosylation, type II; IDD, X-linked syndromic, Hedera type	GTCS, atonic, myoclonic	4–14 mo	NR		Generalized polyspike and waves discharges before seizures	NR	Cerebellar atrophy and/or thinning of the CC	Scoliosis, ataxia and hyporeflexia, Parkinsonism	NR		Hedera, 2002 [35]; Ramser, 2005 [36]; Gupta, 2015 [37]
*IQSEC2*	Xp11.22	XLD		IDD, X-linked 1	Atonic, myoclonic, IS atypical absences, GTCS	8 mo–4 Y	NR	Not specific		NR				Yes	Zerem, 2016 [38]; Choi, 2020 [39]
*NEXMIF*	Xq13.2	XLD		IDD X-linked 98	EEM, EMAtS	NR			Generalized SW or polyspikes Photo-sensitivity, ECS	ID, ASD	Normal	Hypotonia, ataxia, microcephaly, gastro-esophageal reflux disease, strabismus, dysmorphic features	NR	NR	Stamberger, 2021 [40]; Coppola, 2023 [41]
*PIGA*	Xp22.2	XLR		Multiple congenital anomalies-hypotonia-seizures syndrome; Neurodevelopmental disorder with epilepsy and hemochromatosis; Paroxysmal nocturnal hemoglobinuria	Myoclonic, tonic S with apnea, GTCD, Focal S, migrating focal S, atonic, gelastic, IS, FS	Firs Y		Diffuse background slowing, multifocal or diffuse spike/sharp and SW with high amplitude	NR	PMD, ID	Cortical and subcortical volume loss, cerebellar and/or brainstem hypoplasia, white matter immaturity, corpus callosum dysgenesis; FCD, HS	Congenital heart anomalies	KD, pyridoxine	Yes	Kim, 2016 [42]; Bayat, 2020 [43]; Ikeda, 2023 [44]
*ALG13*	Xq23	XL		DEE 36	IS, tonic S, LGS	6 mo	NR	hypsarrythmic pattern	paroxysms of fast activity with electrodecrement without clinical correlate	PMD with regression	matter volume loss, corpus callosum dysgenesis and cortical atrophy	hypotonia, movement disorders, dysmorphic features.	NR	yes	Datta, 2020 [45]; Berry, 2020 [46]
*FHF2/* *FGF13*	Xq26.3-q27.1	XLD, XLR		DEE 90	motor features, apneas, oro-alimentary automatism, IS, GTCS, GEFS+	NR	NR	temporal focus	NR	NR	NR	NR	Surgery	yes	Fry, 2021 [47]; Narayanan, 2022 [48]
*GRIA3*	Xq25	XLR		ID, X-linked syndromic, Wu type	Myoclonic S, is, atypical absences	First mo	NCSE	Multifocal and generalized abnormalities; subcontinuous diffuse epileptiform abnormalities with higher amplitude over bilateral frontal regions	NR	ID, ASD	NR	hypotonia, asthenic body habitus with poor muscle bulk, hyporeflexia.	PER	NR	Trivisano, 2020 [49]; Sun, 2021 [50]
*SMC1A*	Xp11.22	XLD		Cornelia de Lange syndrome 2; DEE 85, with or without midline brain defects	clustering seizures	NR	NR	NR	NR	ID	NR	NR	NR	Yes	Selicorni, 2021 [51]; Symonds, 2017 [52]; Bozarth, 2023 [53]

Legend: ASD: autism spectrum disorder; BDZ: Benzodiazepine; BPNH: Bilateral periventricular nodular heterotopia; CC: corpus callosum; DEE: developmental and epileptic encephalopathy; DRE: drug resistance epilepsy; ECS: eye closure sensitivity; EEM: eyelid myoclonia; EMAtS: epilepsy with myoclonic–atonic seizures; FCD: focal cortical dysplasia; FS: febrile seizures; GTCS: generalized tonic–clonic seizure; GEFS+: Genetic Epilepsy with Febrile Seizures Plus; HS: hippocampal sclerosis; ID: intellectual disabilities; IDD: Intellectual developmental Disorder; IS: infantile spasm; LGS: Lennox-Gastaut syndrome; Mo: months; NCSE: Non Convulsive Status Epilepticus; NR: not reported; PER: Perampanel; PMD: psycho-motor delay; S: seizures; SB: suppression burst; SBH: Subcortical band heterotopia; SE: Status Epilepticus; SMo: somatic mosaicism; SW: slow waves; XLD: X-linked dominant; XLR: X-linked recessive; Y: year.

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
