# Peer review of "X-Linked Epilepsies: A Narrative Review"

_ijms, 2024, doi:10.3390/ijms25074110_

Round 1

Reviewer 1 Report

Comments and Suggestions for Authors

In the manuscript X-linked epilepsies: a narrative review” by Drs. Pia Bernardo et al authors reviewed the main features of  X-linked epileptic syndromes thoroughly characterized to date (PCDH19-related DEE, CDKL5-related DEE, MECP2-related disorders), forms of epilepsy related to X-linked neuronal

migration disorders (ARX, DCX, FLNA) and developmental and epileptic encephalopathies (DEEs) associated with many genes including SLC9A6, SLC35A2, SYN1, ARHGEF9, ATP6AP2, IQSEC2, NEXMIF, PIGA, ALG13, FGF13, GRIA3, SMC1A). Definitely, different modes of X-linked inheritance are possible and modifying factors, including epigenetic regulation and X-chromosome inactivation and the authors was trying to guide the clinician in the genetic diagnosis and treatment of patients with epilepsy featuring X-linked inheritance.

 The authors have done a great job and systematized extensive material. I have no substantive objections, but there are some questions that I would like to clarify.

How do the data on epileptic activity of carriers of a mutation in the MECP2 gene correspond to the data obtained in animal models? It has been shown that in male Mecp2-deficient mice, the activity of neurons in the somatosensory cortex evoked by whisker stimulation has significant differences. There are also differences in the histological structure of layer IV of the cortex. Li-Jen Lee et al, J Comp Neurol, doi: 10.1002/cne.24315. I would advise using this publication and perhaps some others to shed light on this issue.

The authors use the term "psychomotor arrest" - this term is often used but its meaning is not well defined. It makes sense to add a few words to clarify its meaning in the context in which the authors use it

The authors mention Christianson syndrome associated with the SLC9A6 gene. From my point of view, important data on this gene and the ion channel it encodes were obtained recently from experiments with mutant mice. Authors can easily find this publication; perhaps it will strengthen the manuscript.

The material is adequate to the task, and the review of the data obtained is beyond doubt. I will be happy to recommend the manuscript for publication after the corrections indicated above.

Author Response

Dear Editors,

Please, find enclosed a revised version of our manuscript titled "X-linked epilepsies: a narrative review". We thank the Reviewers for their thorough evaluation and constructive comments.

We are grateful for the opportunity to address these comments and clarify certain aspects of our narrative review. We went through the manuscript carefully and have prepared a detailed response to the Reviewers.

Reviewer #1: In the manuscript “X-linked epilepsies: a narrative review” by Drs. Pia Bernardo et al authors reviewed the main features of X-linked epileptic syndromes thoroughly characterized to date (PCDH19-related DEE, CDKL5-related DEE, MECP2-related disorders), forms of epilepsy related to X-linked neuronal migration disorders (ARX, DCX, FLNA) and developmental and epileptic encephalopathies (DEEs) associated with many genes including SLC9A6, SLC35A2, SYN1, ARHGEF9, ATP6AP2, IQSEC2, NEXMIF, PIGA, ALG13, FGF13, GRIA3, SMC1A). Definitely, different modes of X-linked inheritance are possible and modifying factors, including epigenetic regulation and X-chromosome inactivation and the authors was trying to guide the clinician in the genetic diagnosis and treatment of patients with epilepsy featuring X-linked inheritance.

The authors have done a great job and systematized extensive material. I have no substantive objections, but there are some questions that I would like to clarify.

How do the data on epileptic activity of carriers of a mutation in the MECP2 gene correspond to the data obtained in animal models? It has been shown that in male Mecp2-deficient mice, the activity of neurons in the somatosensory cortex evoked by whisker stimulation has significant differences. There are also differences in the histological structure of layer IV of the cortex. Li-Jen Lee et al, J Comp Neurol, doi: 10.1002/cne.24315. I would advise using this publication and perhaps some others to shed light on this issue.

We thank the Reviewer for the kind comment and hope we have addressed his/her concerns in the revised version of the manuscript. We added the suggested reference in the paragraph “4.3.1. Rett syndrome (RTT)”, page 12, line 200-204 of the untracked manuscript: “Moreover, in RTT mouse models, significant structural and functional impairments affecting the thalamo-cortical circuitry and the developing somatosensory cortex have been observed. These anomalies have been hypothesized to contribute to the sensorimotor stereotypies characteristic of RTT”.

The authors use the term "psychomotor arrest" - this term is often used but its meaning is not well defined. It makes sense to add a few words to clarify its meaning in the context in which the authors use it.

We thank the Reviewer for his/her correct clarification. We have replaced the term "psychomotor arrest" with "behavior arrest” in Figure 2 Legend, page 9, line 99 of the untracked manuscript. According to the basic ILAE 2017 operational classification of seizure types, behavior arrest is a non-motor seizure type and consists of cessation of activity (Fisher RS et al. Epilepsia. 2017 Apr;58(4):522-530. doi: 10.1111/epi.13670).

The authors mention Christianson syndrome associated with the SLC9A6 gene. From my point of view, important data on this gene and the ion channel it encodes were obtained recently from experiments with mutant mice. Authors can easily find this publication; perhaps it will strengthen the manuscript.

We thank the reviewer for the valuable suggestion, and we added some information about Christianson syndrome pathobiology in the paragraph “6.1. SLC9A6”, page 17, lines 414-415 of the untracked manuscript: “The NHE6 is involved in many cellular processes critical to normal nervous system development and responsiveness to noxious stimuli”.

The material is adequate to the task, and the review of the data obtained is beyond doubt. I will be happy to recommend the manuscript for publication after the corrections indicated above.

We thank the Reviewer for the suggestions, and we hope to have addressed his/her concerns.

Reviewer 2 Report

Comments and Suggestions for Authors

To the AA

X-linked epilepsies (XLE) are reviewed in the present Ms. Details for several XLE are reported. Methodology was exclusively based on MEDLINE PubMed search. Bibliography contains a list of 136 Refs. The AA’s conclusions are towards a possible help to clinicians in the diagnosis and treatment of patients with suspected XLE.

Major Point

XLE are really of increasing interest to pediatricians, geneticists and neurologists as well.  The review fails to go beyond a mere list of reports. Used methodology has intrinsic limitations, being very little data originating from the AA’s own clinical experience. The latter is mainly expressed as polygraphic recordings in single case-reports and it regards just a minority of the described XLE conditions.  Unfortunately, for several of the listed conditions, a really systematic information is rarely available on the published literature. Apparently, only 17% of the cited Refs (23 out 136 Refs) were published in the last 5 years (years 2020 to 2024), and less than 5% (5 out of 136) of the Refs include at least one of the AA of the present Ms. Therefore, and I cannot understand how this narrative review would contribute to the diagnosis and treatment of patients with suspected XLE. For considering the Ms suitable for publication, the AA should at least convincingly support the contribution to the field provided by the present Ms.

Minor Points

1)      A table with a description of the known XLE prevalences could be of help for the potential clinical readership;

2)      A table with the number and characteristics of the XLE case-series personally evaluated and treated by the AAs;

3)      Refs format is sometimes not coherent, and should be corrected;

4)      Figure 1 is not necessary, as it does not convey useful information besides basic knowledge;

5)      Minor to moderate language editing is needed

Comments on the Quality of English Language

1   Minor to moderate language editing is needed

Author Response

Dear Editors,

Please, find enclosed a revised version of our manuscript titled "X-linked epilepsies: a narrative review". We thank the Reviewers for their thorough evaluation and constructive comments.

We are grateful for the opportunity to address these comments and clarify certain aspects of our narrative review. We went through the manuscript carefully and have prepared a detailed response to the Reviewers.

Reviewer #2: To the AA: X-linked epilepsies (XLE) are reviewed in the present Ms. Details for several XLE are reported. Methodology was exclusively based on MEDLINE PubMed search. Bibliography contains a list of 136 Refs. The AA’s conclusions are towards a possible help to clinicians in the diagnosis and treatment of patients with suspected XLE.

Major Point

XLE are really of increasing interest to pediatricians, geneticists and neurologists as well.  The review fails to go beyond a mere list of reports. Used methodology has intrinsic limitations, being very little data originating from the AA’s own clinical experience. The latter is mainly expressed as polygraphic recordings in single case-reports and it regards just a minority of the described XLE conditions.  Unfortunately, for several of the listed conditions, a really systematic information is rarely available on the published literature. Apparently, only 17% of the cited Refs (23 out 136 Refs) were published in the last 5 years (years 2020 to 2024), and less than 5% (5 out of 136) of the Refs include at least one of the AA of the present Ms. Therefore, and I cannot understand how this narrative review would contribute to the diagnosis and treatment of patients with suspected XLE. For considering the Ms suitable for publication, the AA should at least convincingly support the contribution to the field provided by the present Ms.

We thank the reviewer for his/her concerns. We believe this comment helps us improving our work. XLE include a large and expanding group of epileptic syndromes and/or developmental and epileptic encephalopathies. Very few reviews have been provided in this regard and mainly include the forms best detailed in recent years (e.g. CDKL5-DEE, PCDH19-DEE) or the well-known Rett syndrome. Other genes responsible for XLEs have been identified very recently and less clinical data have been described in literature. Furthermore, these conditions are very rare and as such our personal experience does not cover all the described X-linked epilepsies. Moreover, the aim of this work is to provide a detailed narrative review and not an original article reporting our own experience. However, please note that we have a good and direct (clinical, electrophysiological and genetic) experience with the most common XLE including MECP2, CDKL5, NEXMIF, MECP2 and ARX related epilepsies as demonstrated by the following manuscripts we have published:  

  1. Lotte J, Bast T, Borusiak P, Coppola A, Cross JH, Dimova P, Fogarasi A, Graneß I, Guerrini R, Hjalgrim H, Keimer R, Korff CM, Kurlemann G, Leiz S, Linder-Lucht M, Loddenkemper T, Makowski C, Mühe C, Nicolai J, Nikanorova M, Pellacani S, Philip S, Ruf S, Sánchez Fernández I, Schlachter K, Striano P, Sukhudyan B, Valcheva D, Vermeulen RJ, Weisbrod T, Wilken B, Wolf P, Kluger G. Effectiveness of antiepileptic therapy in patients with PCDH19 mutations. Seizure. 2016 Feb;35:106-10. doi: 10.1016/j.seizure.2016.01.006. Epub 2016 Jan 6.
  2. Bernardo P, Ferretti A, Terrone G, Santoro C, Bravaccio C, Striano S, Coppola A, Striano P. Clinical evolution and epilepsy outcome in three patients with CDKL5-related developmental encephalopathy. Epileptic Disord. 2019 Jun 1;21(3):271-277. doi: 10.1684/epd.2019.1071.
  3. Bernardo P, Cobb S, Coppola A, Tomasevic L, Di Lazzaro V, Bravaccio C, Manganelli F, Dubbioso R. Neurophysiological Signatures of Motor Impairment in Patients with Rett Syndrome. Ann Neurol. 2020 May;87(5):763-773. doi: 10.1002/ana.25712. Epub 2020 Mar 17.
  4. Bernardo P, Coppola A, Terrone G, Riccio MP, Santoro C, Del Giudice E, Bravaccio C. Epilepsy in Rett Syndrome: can seizures play an encephalopathic effect in this disorder? Minerva Pediatr. 2019 Aug;71(4):391-393. doi: 10.23736/S0026-4946.19.05309-X. Epub 2019 Apr 5.
  5. Cappuccio G, Bernardo P, Raiano E, Pinelli M, Alagia M, Esposito M, Della Casa R, Strisciuglio P, Brunetti-Pierri N, Bravaccio C. Pain and sleep disturbances in Rett syndrome and other neurodevelopmental disorders. Acta Paediatr. 2019 Jan;108(1):171-172. doi: 10.1111/apa.14576. Epub 2018 Oct 17.
  6. Bernardo P, Raiano E, Cappuccio G, Dubbioso R, Bravaccio C, Vergara E, Peluso S, Manganelli F, Esposito M. The Treatment of Hypersalivation in Rett Syndrome with Botulinum Toxin: Efficacy and Clinical Implications. Neurol Ther. 2019 Jun;8(1):155-160. doi: 10.1007/s40120-018-0125-9. Epub 2019 Jan 8.
  7. Drongitis D, Caterino M, Verrillo L, Santonicola P, Costanzo M, Poeta L, Attianese B, Barra A, Terrone G, Lioi MB, Paladino S, Di Schiavi E, Costa V, Ruoppolo M, Miano MG. Deregulation of microtubule organization and RNA metabolism in Arx models for lissencephaly and developmental epileptic encephalopathy. Hum Mol Genet. 2022 Jun 4;31(11):1884-1908. doi: 10.1093/hmg/ddac028.
  8. Coppola A, Krithika S, Iacomino M, Bobbili D, Balestrini S, Bagnasco I, Bilo L, Buti D, Casellato S, Cuccurullo C, Ferlazzo E, Leu C, Giordano L, Gobbi G, Hernandez-Hernandez L, Lench N, Martins H, Meletti S, Messana T, Nigro V, Pinelli M, Pippucci T, Bellampalli R, Salis B, Sofia V, Striano P, Striano S, Tassi L, Vignoli A, Vaudano AE, Viri M, Scheffer IE, May P, Zara F, Sisodiya SM. Dissecting genetics of spectrum of epilepsies with eyelid myoclonia by exome sequencing. 2023 Dec 13. doi: 10.1111/epi.17859. Epub ahead of print.
  9. Nissenkorn A, Kluger G, Schubert-Bast S, Bayat A, Bobylova M, Bonanni P, Ceulemans B, Coppola A, Di Bonaventura C, Feucht M, Fuchs A, Gröppel G, Heimer G, Herdt B, Kulikova S, Mukhin K, Nicassio S, Orsini A, Panagiotou M, Pringsheim M, Puest B, Pylaeva O, Ramantani G, Tsekoura M, Ricciardelli P, Lerman Sagie T, Stark B, Striano P, van Baalen A, De Wachter M, Cerulli Irelli E, Cuccurullo C, von Stülpnagel C, Russo A. Perampanel as precision therapy in rare genetic epilepsies. Epilepsia. 2023 Apr;64(4):866-874. doi: 10.1111/epi.17530. Epub 2023 Feb 20.
  10. Cioclu MC, Coppola A, Tondelli M, Vaudano AE, Giovannini G, Krithika S, Iacomino M, Zara F, Sisodiya SM, Meletti S. Cortical and Subcortical Network Dysfunction in a Female Patient With NEXMIF Encephalopathy. Front Neurol. 2021 Sep 9;12:722664. doi: 10.3389/fneur.2021.722664.

Indeed, we have edited the manuscript including all our previously published works on the topic, as it follows:

  • Paragraph « 4.3.1. Rett syndrome (RTT) », page 12, lines 195-200 of the untracked manuscript: « Motor impairment constitutes one of the core features of RTT, including the development of gait abnormalities, stereotypic hand movement and the progressive deterioration of motor abilities. Impairment of GABAergic inhibitory circuits within the primary motor cortex has been observed in RTT patients through transcranial magnetic stimulation (TMS) protocols and it has been hypothesized as one of the underlying mechanisms of motor dysfunction ».
  • Paragraph « 4.3.1. Rett syndrome (RTT) », page 13, lines 241-243 of the untracked manuscript: « To conclude, additional features may be observed in RTT individuals, partly included within the supportive criteria, as impaired nociception, sleep disturbances, hypersalivation due to oral motor dysfunction ».
  • Paragraph « 5.1 ARX », page 16, lines 342-343 of the untracked manuscript: « Pathogenic variants in ARX lead to functional alterations affecting microtubule network and RNA metabolism».   
  • Paragraph « 6.7 NEXMIF», page 19, lines 498-502 of the untracked manuscript: « Advanced neuroimaging techniques have revealed brain structural and functional changes especially in the visual pericalcarine cortex and in the middle frontal gyrus, which are involved in several visuomotor function and therefore in epilepsy phenotypes with visual sensitivity and eyelid myoclonia ».

Regarding the provided illustrative images, these are not yet published polygraphic EEG recordings belonging indeed to patients followed up at our epilepsy centers.

We agree that being this work a narrative review this cannot be considered a guide to help clinicians diagnose and treat XLE. We have toned this down and clarified the purpose of this updated narrative review both in the Abstract, page 1, lines 29-30 of the untracked manuscript and in the paragraph “Conclusion”.

We hope that this review will increase attention to suspect X-linked inheritance in some forms of epilepsy.

Minor Points

1)      A table with a description of the known XLE prevalences could be of help for the potential clinical readership;

We thank the reviewer for the valuable suggestion. We have added the known incidence of some XLE in Table 1. The prevalence and incidence of the remaining XLE genes variants is difficult to assess and often not reported in literature. This is mostly due to the rarity of the conditions. Also, individuals with mild phenotype may never seek medical evaluation (e.g. FLNA).

2)      A table with the number and characteristics of the XLE case-series personally evaluated and treated by the AAs;

We have not provided a description of our case series as this was beyond the scope of the present work. This work was conceived as an updated narrative review of the literature, particularly focused on the electro-clinical characteristics of both known and emerging XLEs. As these conditions are rare, our personal experience would not cover all of them.  

Here is a table of the individuals followed in our Epilepsy Centers. We would not add this table as part of the manuscript. We suggest adding this as supplementary table. However, we would be happy to adjust our thought as required by the editors. And we are willing to implement the information extensively if required.

Gene

#patients

Phenotype

PCDH19

8 patients

Typical PCDH19-DEE with Clusters of FS, focal S, hypomotor and affective seizures

CDKL5

3 patients harboring de novo variants

1 patient with Xp22 deletion (13.804 Kb) including CDKL5

Tonic-spasms during sleep

Drug-resistant epileptic encephalopathy (prevalent tonic seizures)

MECP2

72 patients

Both focal and generalized epilepsy (15/72 drug-resistant; about 21%)

MECP 2 duplication

2 males and 1 female

2 (1F, 1M) Lennox Gastaut Syndrome

 1M with Lennox-like phenotype (late-onset drug-resistant epileptic spasms)

FLNA

2 patients

Focal epilepsy, drug responsive

IQSEC2

1 patient

Generalized epilepsy (tonic-clonic and atonic seizures)

NEXMIF

2 patients

Epilepsy with eyelid myoclonia and Intellectual disability

Focal epilepsy with ID

3)      Refs format is sometimes not coherent, and should be corrected;

We thank the reviewer for this observation. We have corrected the references format.

4)      Figure 1 is not necessary, as it does not convey useful information besides basic knowledge;

We thank the reviewer for this consideration. We have prepared Figure 1 in order to illustrate the variable modes of X-linked inheritance and additional modifying factors. We insist on believing that this figure may be of help for the readers as it shows the different modes of X-linked inheritance and helps understanding the complexity of genotype-phenotype correlations in XLEs. We would like to keep it. However, if the journal requires to sacrifice it, we will of course agree.

5)      Minor to moderate language editing is needed

We have revised the manuscript and corrected typos and performed extensive language editing. All the changes are tracked.

We believe these revisions have substantially improved our manuscript by clarifying these crucial points and addressing the reviewers' concerns. We deeply thank the reviewers for their valuable comments indeed. We appreciate the opportunity to enhance our work and hope the revisions meet the reviewers' and editorial board's expectations.

Please note that in the revised version of the manuscript Professor Taglialatela requested to be removed as an author. We have attached his signed letter.

Thank you for considering our revised submission for publication in International Journal of Molecular Sciences. We remain at your disposal for any further information or clarification required.

Round 2

Reviewer 2 Report

Comments and Suggestions for Authors

This reviewer really appreciates the efforts made by the AAs in addressing the major and minor points of concern / potential improvement. Organization and readability of the revised Ms. are certainly improved. 

Comments on the Quality of English Language

Minor editing is required. Nonetheless, this could safely be left to the proof correction phase (provided that the Ms will be accepted by the Editor/s)